ⓐ | **Open Peer Review** | Host-Microbial Interactions | Minireview

# Heterogeneity of bacterial host-pathogen interactions across biological scales

**Juan Alfonso Redondo,[1] Niharika Purkait,[1] Pawel Paszek[1,2,3]**

**ABSTRACT** Recent advances in single-cell technologies have revealed the dynamic and heterogeneous nature of host-pathogen interactions at the single-cell level. This review explores how cellular variability—both within clonal bacterial populations and among genetically identical host cells—gives rise to distinct infection outcomes, from pathogen clearance to persistence across multiple biological scales, from single cells to tissues and the whole organism. We highlight the conceptual and technological progress that has enabled the dissection of these interactions at single-cell resolution, including microscopy, single-cell transcriptomics, proteomics, and emerging dual RNA-seq and spatial approaches. Drawing on examples from well-characterized bacterial pathogens like *Listeria monocytogenes*, *Salmonella enterica,* and *Mycobacterium tuberculosis*, we discuss how stochastic gene expression, intrinsic and extrinsic factors, as well as tissue context shape the variable activation of the immune responses and ultimately determine the outcomes of host-pathogen interactions. We argue that the outcome of single-cell interactions is shaped by a combination of host states, bacterial-intrinsic features, and the local microenvironment. We further discuss how computational and mathematical modeling can integrate these heterogeneous single-cell events across spatial scales, linking intracellular variability with tissue-level pathogenesis and progression of infection. Gaining insight into and controlling these layers of variability holds promise for the development of more precise, context-dependent antimicrobial strategies.

**KEYWORDS** host-pathogen interactions, single-cell biology, cellular heterogeneity, infection biology, *Listeria monocytogenes*, *Salmonella enterica*, *Mycobacterium tuberculosis*

Infection biology has traditionally relied on bulk assays to measure average host responses and pathogen behavior. However, advances in single-cell technologies are now revealing that infection is a heterogeneous process, governed by seemingly probabilistic and dynamic interactions resulting in different infection outcomes.

Genetically identical host cells can differ in susceptibility to infection due to variations in their receptor abundance, transcriptional state, or global cellular processes, like, for example, cell cycle (1). Clonal bacterial populations exhibit phenotypic variability in the expression of virulence genes, stress resistance pathways, and surface antigens (2). These differences, intrinsic or environmentally induced, can persist for several cell divisions and lead to functional heterogeneity of cell states and fates, ultimately deciding infection success or failure (3–6). For example, a subset of host cells may initiate productive inflammatory responses, whereas others support bacterial replication and spread or serve as niches for persistence (7–9). These rare events can disproportionately shape the course of infection (10). Tissue-specific organization, immune dynamics (11), and pathogen phenotypic heterogeneity (12) contribute to disease progression, immune evasion, and therapeutic response, demonstrating how heterogeneous single-cell events can have functional consequences at tissue and whole organism levels (Fig. 1).

**Peer Reviewer** Bingqing Li, Shandong Academy of Medical Sciences, Jinan, Shandong, China

Address correspondence to Pawel Paszek, ppaszek@ippt.pan.pl.

The authors declare no conflict of interest.

See the funding table on p. 14.

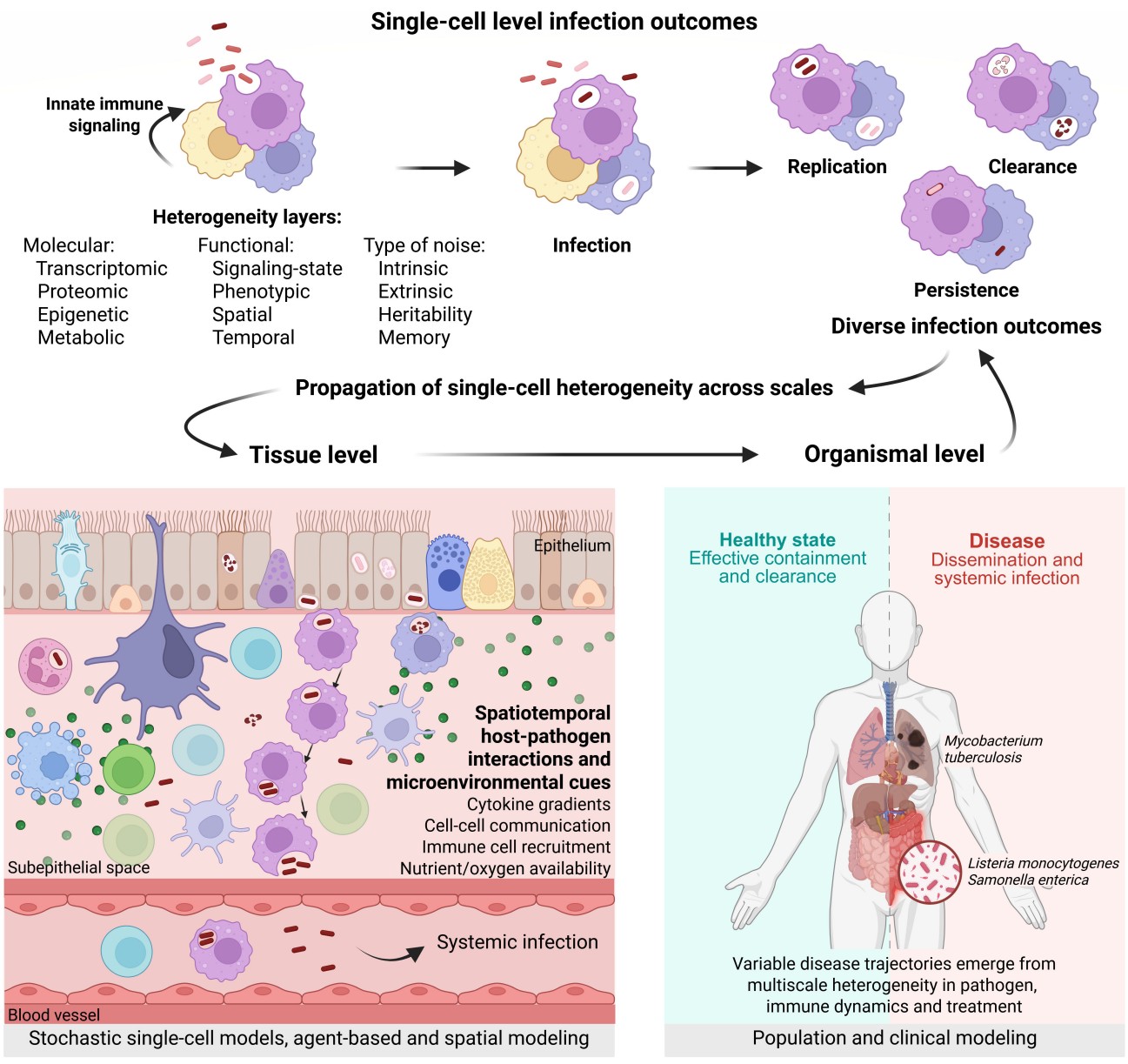

**FIG 1** Heterogeneous nature of host-pathogen interactions across scales. Cellular variability arising from different layers of regulation influences single-cell host-pathogen interactions and, ultimately, infection outcomes. This heterogeneity propagates across scales from individual cells to tissue and organismal level and can be better understood by using mathematical modeling approaches.

In the present review, we argue that cellular heterogeneity is a mechanistic driver of infection outcomes. We integrate technological, biological, and modeling perspectives to outline how single-cell variability, in both host and pathogen, translates to tissue-level immunity and disease and paves the way to therapeutic manipulation in a more precise, context-dependent manner.

## MEASUREMENT OF HOST-PATHOGEN INTERACTIONS AT THE SINGLE-CELL LEVEL

Single-cell technologies have transformed our ability to dissect host-pathogen interactions at the single-cell level. However, as the variability in the host and pathogen is interdependent, neither side can be understood in isolation; simultaneous measurement is therefore required (13). Below, we outline how different single-cell approaches

differ in the number and type of readouts they provide, their spatial and temporal resolution, and how these features critically affect their ability to quantify host and pathogen behaviors within the same experiment (Table 1).

## Microscopy approaches for quantification of interactions

Microscopy has remained a cornerstone of infection biology for hundreds of years, providing quantitative data with high spatial and temporal resolution. In particular, live-cell microscopy is uniquely suited to monitor heterogeneity in host and pathogens at the molecular, cellular, and population/organ scales. It enables direct observation of infection dynamics, including bacterial entry, intracellular growth, virulence activation, and host signaling responses, making it possible to connect differences in host-cell state to divergent pathogen behaviors in real time (8, 14–16). High-content microscopy further scales these measurements, providing automated imaging of hundreds of fields or wells with tens of thousands of cells (17). This high-content microscopy can be combined with AI-based image segmentation and classification tools, allowing the extraction of key infection features such as pathogen load, subcellular localization, and host response markers at single-cell resolution (18). Deep-learning models now reliably segment bacteria within complex host environments and detect rare infection outcomes, outperforming classical threshold-based pipelines under variable imaging conditions (19–23). However, even with advanced segmentation, most pipelines still have limited capacity to integrate multi-dimensional host data (e.g., signaling state and metabolic status) with equally rich pathogen-level features, making it difficult to reconstruct or interpret their interactions.

Advanced imaging modalities further expand the mechanistic scope of microscopy (24). Beyond static visualization, techniques typically applied to live-cell imaging—such as Förster resonance energy transfer (FRET) and fluorescent recovery after photobleaching (FRAP)—quantify molecular interactions and protein mobility inside host cells (25,

**TABLE 1** Single-cell biology approaches for host-pathogen interactions

| Technology | Throughput (number of targets) | Spatial/temporal resolution | Dual host and pathogen detection | Strengths | Limitations |
|---|---|---|---|---|---|
| Fluorescent microscopy | Live-cell: 3–4 probes; fixed-cell: >100 via multiplexing | Real-time, spatial, subcellular | Yes | Direct kinetic insights, visual observation, absolute quantification | Live cell: limited number of processes simultaneously monitored |
| High-content automated microscopy | Hundreds of wells/ fields | Real-time, spatial, subcellular, | Yes | Large-scale screening, feature extraction from images | Computationally intensive; segmentation still imperfect |
| CyTOF/Imaging mass cytometry (IMC) | ~40 protein markers per cell | No temporal, spatial (IMC) | Yes | Single-cell resolution, high multiplexing (IMC) | Antibody-dependent; throughput and cost limitations |
| Label-free single-cell MS proteomics | ~1,000 proteins per cell | No | Mostly host; pathogen proteins underrepresented | Unbiased proteome interrogation | Low sensitivity in bacteria; limited throughput |
| scRNA-seq | >10,000 genes per cell | Very limited temporal resolution | Mostly host; limited pathogen with custom protocols | High multiplex transcriptomic; assay host variability | Commercial protocols exclude non-polyA transcripts |
| Spatial transcriptomics | 1,000s of transcripts per spot/pixel | 1–10 µm for host, 50 µm for pathogen reads | Mostly host; pathogen detection possible | Preserves tissue architecture | Typically, low read depth; limited pathogen RNA detection |
| Single-cell epigenomics | Genome-wide regulatory profiles | Spatial (>10 µm in the host) | Both, but not simultaneously | Reveals chromatin state, can be multiplexed with RNA | Analyzes limited in bacteria |
| Dual scRNA-seq | ~10,000 host reads; 100–300 bacterial genes per cell | No | Mostly host; limited detection of pathogen | Correlation between host and pathogen responses | Low pathogen depth; complex protocols; low throughput |

26). Complementarily, live-cell RNA tracking (27, 28) can, in principle, reveal how transcriptional dynamics unfold during infection. However, these techniques often preferentially capture either host or pathogen readouts, limiting their ability to resolve their reciprocal relationship. Moreover, live imaging remains restricted to ~4 channels due to fluorophore spectral overlap, phototoxicity, and limited signal amplification (29). Fixed-cell multiplexing technologies overcome these barriers, enabling detection of hundreds of proteins and thousands of mRNAs *in situ* (30–32), and providing absolute transcript counts, capturing cell-to-cell variability in mRNA abundance. These fixed-cell methods greatly improve analytical throughput but do not preserve the temporal relationship between pathogen and host responses.

Overall, microscopy provides rich quantitative and mechanistic insights but is limited by the tradeoff between temporal resolution and depth of the analyses. Consequently, resolving host and pathogen heterogeneity often requires the integration of microscopy with single-cell transcriptomic or proteomic approaches.

## Proteomics for single-cell host-pathogen interactions

Proteomic approaches have traditionally relied on antibody-based detection, with flow cytometry quantifying over 20 proteins per cell for traditional infection phenotyping (33). However, imaging flow cytometry (IFC) platforms combine cytometry with imaging to enable robust intracellular pathogen detection and analyses of pathogen morphological features relevant to intracellular infection stages, beyond protein expression (34).

Mass cytometry (CyTOF) extends flow cytometry's multiplexing capability by using metal-tagged antibodies, allowing the detection of ~40 proteins per cell (35) and enabling dual detection of host and pathogen antigens (36). Spatial mass cytometry (IMC) further adds tissue context, mapping protein expression within intact infected microenvironments (37). Although mass cytometry can measure host-pathogen components concurrently (38), throughput is limited and dependent on validated antibody panels, especially on the pathogen side.

Label-free single-cell mass spectrometry (scMS) bypasses antibody constraints by directly quantifying peptides. Current methods still detect only the most abundant ~1,000 proteins per cell, despite advances in the sensitivity of the proteome coverage (39, 40). Low-abundant pathogen proteins remain difficult to capture, as they are vastly outnumbered by host peptides, thus limiting their utility for dual host-pathogen analyses. Population-level bulk proteomics has routinely detected bacterial proteins in infected cells (41), indicating that improved sensitivity may eventually enable true dual-proteome single-cell measurements.

Overall, single-cell proteomics provides robust and highly multiplexed readouts of host responses, but simultaneous host-pathogen proteome measurements remain limited by pathogen protein abundance and instrument sensitivity.

## Measurement of transcriptional variability in both host and pathogen

Single-cell transcriptomics and genomics have revolutionized the study of infection biology by enabling unbiased, genome-wide profiling of host responses with single-cell, spatial (42), and even temporal resolution (43). Among different approaches, single-cell RNA sequencing (scRNA-seq) has become a key tool for revealing variability in transcriptomic responses to infection in the host (13, 44–47) and bacterial populations (48), laying the foundation for spatial transcriptomic approaches that map host-pathogen interactions directly within tissues (49, 50). Single-cell epigenomics allows the detection of an additional regulatory layer, revealing expression programs at the level of chromatin and DNA availability. In particular, single-cell ATAC-seq (scATAC-seq) (51) has been used to chart the host chromatin landscape during infection and vaccination (52, 53). Emerging technologies like CUT&Tag (54) and scChiC-seq (55) enable the profiling of histone marks and transcription factor occupancy, while multimodal methods combine chromatin accessibility, transcriptomics, and protein abundance within the same single host cells (56–58).

Despite its transformative potential, single-cell transcriptomics faces several barriers in dual host-pathogen profiling, particularly on the pathogen side (48). The minute pathogen RNA content per cell requires extensive amplification, biasing detection toward abundant transcripts and limiting recovery of low-copy bacterial RNAs (59). A further challenge is that most commercial single-cell protocols, such as Smart-seq2 and 10× Chromium, rely on oligo(dT) priming, systematically preventing capture of poly(A)-independent bacterial RNAs (60), limiting standard scRNA-seq workflows from capturing bacterial RNA.

Dedicated bacterial scRNA-seq protocols are beginning to address these limitations (61–64), paving the way for simultaneous detection of host and pathogen responses. Bulk dual RNA-seq studies have already demonstrated the power of this approach to uncover co-variation between population-level host-pathogen responses, even highlighting the roles of non-coding RNAs (65) and host metabolites (66) in triggering bacterial virulence programs. Extending dual RNA-seq to the single-cell level has proved to be challenging. Avital et al. provided proof of principle by simultaneously profiling macrophages infected with *Salmonella enterica* (*S. enterica*) using a custom plate-based scDual-Seq approach. They used random hexamer primers to reverse-transcribe both host and bacterial RNAs, followed by artificial polyadenylation for standard library construction (67). At high multiplicity of infection (MOI = 50), they detected a median of 10,000 *Salmonella* reads and 200–300 bacterial genes per cell. Building on this, Heyman et al. developed scPAIR-seq, a single-cell screening platform where pooled *Salmonella* mutants express unique polyadenylated barcodes captured alongside the host transcriptome, providing genotype-phenotype resolution even without bacterial transcript capture (68).

Dual scRNA-seq has also been applied to *Leishmania donovani* infection *in vivo*. Despite the limited coverage of the 10× Genomics platform, residual polyadenylated parasite transcripts enabled inference of rare infected cell types (69). The utility of these low-abundance reads is highlighted by emerging spatial transcriptomics platforms that capture both host and pathogen transcripts with increasing spatial resolution (70, 71), paving the way toward subcellular resolution (72) and spatial epigenetics (73) as already demonstrated in host tissues. These residual bacterial reads can be computationally recovered from standard scRNA-seq data sets (74), enabling retrospective detection of infection status.

Overall, these single-cell genomics advances highlight the promise and limitations of dual single-cell approaches, where simultaneous host and pathogen measurement will require enhanced amplification strategies, improved bacterial RNA enrichment, and scalable protocols compatible with droplet-based platforms.

## SOURCES OF HETEROGENEITY IN HOST-PATHOGEN INTERACTIONS

Even without infection, genetically identical host cells vary in their immune effector expression and signaling responses. This response variability in single cells often generates cellular heterogeneity, that is, the presence of distinct biological states or cellular subpopulations that differ in fate, or function, for example, the ability to mount a successful immune response. Classical studies established that much of the variability arises from transcription and translation, which generates fluctuations in mRNA and ultimately, protein through an all-or-none, probabilistic gene activation process. Some genes display heritable activation states across consecutive cell divisions, while others show purely stochastic activation with no memory of parental expression (75, 76).

Advances in single-cell technologies have revealed that cellular variability arises from diverse sources. At the single-cell level, intrinsic noise—random fluctuations in promoter binding, transcriptional bursting, and mRNA/protein synthesis/degradation—drives stochastic gene expression patterns due to the low copy number of the molecules involved (77). As a result, most genes are transcribed in random bursts, during brief periods of activity, both in bacteria and mammals (78). Burst size and burst frequency shape patterns of mRNA production in single cells and determine mRNA/protein

distributions across populations (79). The bursting process is dictated by transcription factor signaling, epigenetic regulation, and genome architecture, where core promoters control burst sizes, and enhancers modulate burst frequency (80, 81). In innate immunity, transcriptional bursting governs Toll-like receptor (TLR)- and cytokine-induced gene expression, resulting in critical effector molecules being expressed in only a subset of cells upon stimulation or infection (82, 83). These patterns of stochastic gene expression are not uncontrolled in the conventional sense but follow strict statistical rules. For instance, gene expression noise is inversely proportional to mean expression, while features like switching events, burst sizes, and mRNA/protein levels follow specific distributions that can be theoretically derived (84). This has important implications: TLR-dependent genes display invariant response variability regardless of the stimulus dose- or cellular-context (82, 85). This indicates that variability in the immune responses is an evolutionarily regulated feature of gene expression through genome architecture (46).

Alongside intrinsic noise, extrinsic variability reflects the differences in global factors like signaling, cell size, metabolites, etc., causing coordinated changes across multiple genes (77). Spencer et al. showed that extrinsic variability in protein abundance explained cell-to-cell differences in commitment to apoptosis (86). Extrinsic variability has since been observed in signaling systems, including the immune-related nuclear factor κB (NF-κB), where all-or-none activation reflects differences in receptor and signaling protein levels (87, 88). Cell-to-cell communication, via cytokines and other secreted molecules, is another extrinsic factor shaping response variability in tissue contexts (44). Importantly, extrinsic variability differs from transcriptional heritability or imprinting, which refers to stable gene expression states passed across cell divisions, often via epigenetic modifications or feedback loops (e.g., bistability). For example, early type I interferon (IFN) responsiveness persists in a subset of cells (89), and heritable receptor expression regulates NF-κB activation in daughter cells (88). Finally, transcriptional memory describes a gene's altered responsiveness based on prior activation, often mediated by chromatin accessibility, histone marks, or transcription factor occupancy, for example, by inflammatory signaling (90–92). While classically linked to repeated pathogen exposure, memory-like states may also arise from inherited chromatin configurations (93). Importantly, bacteria also exhibit transcriptional heritability, for example, through epigenetic regulation, where reversible and heritable DNA methylation patterns modulate protein-DNA interactions (94). Classic examples include the pii operon on *Escherichia coli*, which switches between expression states through methylation-dependent Lrp binding, producing heritable phenotypic variation (95).

## Single-cell interactions: Multilayered variability and outcomes

Variability in both host responses and bacterial behavior drives divergent infection outcomes at the single-cell level. Here, we focus on the single-cell interactions of *Listeria monocytogenes*, *Salmonella enterica,* and *Mycobacterium tuberculosis*, three important bacterial pathogens of humans. Despite differences in their invasion mechanisms, recent studies have revealed common factors that shape their interactions within the host (Fig. 1).

### *Listeria monocytogenes*

*Listeria* is a gram-positive foodborne pathogen, responsible for serious infections with up to 30% mortality rate in immunocompromised individuals (96). The potential of *Listeria* to cause systemic infection depends on its ability to cross the intestinal barrier and disseminate in the host (97). Although its intracellular cycle is well established, recent work has revealed the heterogeneous nature of its interactions with host cells.

A key driver of bacterial variability is heterogeneous activation of PrfA, a member of the cyclic AMP (cAMP) receptor family and stress-responsive sigma factor (σB). PrfA directly controls virulence genes required for invasion, vacuolar escape, and actin-based motility, whereas σB governs the general stress response that modulates bacterial fitness

during infection (97). PrfA expression and activity are tightly controlled at multiple levels, including host metabolic cues (98, 99). Under hyperosmotic stress, Utratna et al. showed that only ~60% of *Listeria* induced σB activity (100). Independent fluorescent-reporter studies by Guldimann et al. further showed that PrfA activation is itself variable, with individual bacteria exhibiting broad fluorescence distributions and a larger fraction of cells activating PrfA than σB, indicating distinct activation thresholds (15). During macrophage infection, the PrfA-regulated genes (hly and actA) show marked variability in timing and intensity; replicating bacteria exhibit sustained induction, whereas non-replicating bacteria show weak or transient activity (8). Importantly, constitutive PrfA activation does not increase the proportion of replicating bacteria, indicating that host-cell heterogeneity constrains bacterial fate beyond virulence gene expression.

*Listeria* infection involves attachment, uptake, vacuolar escape, cytosolic replication, and spread (97), with each step being shaped by host-cell heterogeneity. Single-cell imaging reveals a striking variability in phagocytosis, with uptake probabilities ranging from <1% to ~70% depending on cell type and multiplicity of infection (101–103). Variations in scavenger receptor expression and signaling yield fundamentally different entry probabilities across genetically identical macrophages (101, 104, 105). Subsequent vacuolar escape is comparatively efficient (75%–90%) (103, 106, 107); however, intracellular outcomes remain heterogeneous: only ~75% of cytosolic bacteria replicate, whereas the rest enter into non-replicative states (8). Some bacteria also replicate in vacuoles (108), further diversifying bacterial invasion strategies.

In epithelial cells, variability is driven largely by adhesion and receptor availability rather than by internalization or escape (109). Invasion into non-phagocytes is typically rare (<1% even at high MOI) (109–111), but ActA-mediated extracellular aggregation enhances local ligand density and promotes entry (112, 113). Over longer timescales, bacteria can switch between cytosolic motility and vacuolar persistence (5), suggesting that host state modulates bacterial lifestyle choices (99).

Single-cell variability extends to tissue-level dissemination. Rare "pioneer" bacteria travel long distances within cellular layers and determine local infection patterns (114). Spread dynamics also depends on host cytosolic nutrient availability and junctional protein changes (115). Barcoded *Listeria* studies reveal that dissemination to liver and spleen is dominated by a small subset of clones (116, 117), indicating that early variability in cellular encounters determines systemic infection trajectories. In the liver, Kupffer-cell infection and necroptosis trigger differential recruitment and specialization of monocyte-derived macrophages, which together shape tissue-level clearance and resolution (11).

Host defense mechanisms against *Listeria* also exhibit substantial cell-to-cell variability. The extracellular pathogen recognition is mediated by TLR1/2 sensing (118), but TLR expression levels as well as downstream signaling responses exhibit substantial heterogeneity across individual host cells, even in response to uniform synthetic ligand stimulation (88). Sensors of cytosolic bacteria, such as NLRs and cGAS-STING, which have the capacity to directly kill bacteria, are also often activated only in a subset of cells (119–121). Target type I interferons (IFNα/β), induced through NF-κB and IRF3 activation, promote cell-to-cell spread of *Listeria* (122); however, their heterogeneous all-or-none induction patterns may generate local tissue microenvironments that either restrict or facilitate bacterial dissemination. *Listeria* further modulates these pathways through effectors such as InlC and LAP to inhibit or activate NF-κB signaling (123, 124) and exploit tissue- and cell-specific thresholds.

## Salmonella enterica

*S. enterica* is a gram-negative facultative intracellular pathogen comprising over 2,500 distinct serovars. Human-restricted serovars, including *S.* Typhi and *S.* Paratyphi, cause severe systemic infections such as typhoid fever (125). In contrast, non-typhoidal serovars, most notably *S.* Typhimurium, primarily induce acute gastrointestinal inflammation, although in immunocompromised hosts, these infections can progress

to invasive and potentially fatal disease (126). These different clinical outcomes highlight the influence of host immune state, genetic background, and tissue-specific responses in modulating infection trajectories (127).

At the cellular level, *S. enterica* was one of the first bacteria shown to actively invade non-phagocytic cells (128). Invasion is primarily mediated by heterogeneously expressed type III secretion systems (T3SS), particularly SPI-1 and SPI-2. SPI-1 drives the "trigger" entry mechanism involving extensive cytoskeletal reorganization and membrane ruffling, while some serovars use a "zipper-like" mode (129). Once internalized, *S. enterica* resides within the *Salmonella*-containing vacuole (SCV), whose maturation is highly variable due to dynamic interactions with endosomal and Golgi trafficking pathways in the host (130). SPI-1 effectors initiate early CSV formation, while SPI-2 effectors remodel the compartment 2–3 h post-invasion to support bacterial survival (131, 132). In some cells, the SCV membrane is disrupted, allowing cytosolic escape, hyper-replication, inflammasome activation, and epithelial extrusion (133, 134). These intracellular fates of *Salmonella* are extremely heterogeneous: even within the same host cell, genetically identical bacteria may either replicate rapidly, persist slowly, or enter dormancy (135).

Single-cell imaging of fluorescent growth reporters further revealed different and coexisting subpopulations of *Salmonella* in the host, including long-lived persisters and phenotypically diverse vacuolar forms (7, 136). This heterogeneity of bacterial states arises not only from the bacterial genotype but also from host-cell factors, including cell type, activation state, and metabolic state, which affect differences in SCV acidification, lysosomal fusion, ion fluxes, and effector recruitment (137). Bacterial regulatory programs themselves are phenotypically bistable: SPI-1 and SPI-2 gene-expression noise across the population generates bacterial subgroups optimized for invasion or intracellular survival (67, 138), utilizing complex cooperation strategies for colonization (139). Thus, bacterial virulence expression is not uniform but partitioned across subpopulations, seeding heterogeneous infection fates.

Host-cell heterogeneity is equally influential in determining these outcomes. Cell-intrinsic factors, including cell type, polarization state, metabolic activity, and cytoskeletal organization, strongly modulate invasiveness and SCV dynamics (135). For example, the cholesterol content of the host-cell membrane is a major determinant enhancer of bacterial binding. Cholesterol also modulates downstream signaling pathways, like the mammalian mTORC1 pathway and autophagy, ultimately promoting pathogen persistence (140, 141). Furthermore, crowding, membrane tension, and variation in endolysosomal capacity introduce additional layers of heterogeneity at the level of individual host cells (135).

After crossing the gut epithelium, *Salmonella* enters local tissues and organs where it encounters innate immune cells, including monocytes, macrophages, and dendritic cells (142). They display heterogeneity in their basal transcriptional state (44, 46) and differ in their susceptibility to infection (143, 144). Macrophages that support active bacterial replication tend to acquire anti-inflammatory, M2-like transcriptional states, whereas cells harboring non-replicating bacteria preferentially exhibit M1-like, pro-inflammatory programs (45). Additional heterogeneity arises from variation in bacterial virulence traits, such as differential activation of the PhoP-PhoQ two-component system, which alters lipopolysaccharide (LPS) structure and generates a broad range of IFN responses, even among macrophages with an M1-like profile (13). Tissue context amplifies this diversity: in the spleen, CD9$^+$ non-classical monocyte-derived macrophages support *S. Typhimurium* replication, while red-pulp macrophages restrict its growth (145).

### *Mycobacterium tuberculosis*

*M. tuberculosis* (Mtb), the bacterium responsible for tuberculosis (TB), remains one of the leading causes of death from bacterial infection worldwide, disproportionally affecting populations in low- and middle-income countries (146). Infection, most commonly involving the lungs, but Mtb can disseminate systemically, producing a wide range of clinical outcomes from asymptomatic latent infection to progressive, chronic disease

(147). Despite sustained public health interventions and vaccination programs, TB continues to claim more than 1 million lives each year. This ongoing global burden reflects both the emergence of multidrug-resistant strains and pronounced heterogeneity in host immune responses and bacterial pathogenic traits (148).

The foundations of our understanding of Mtb pathogenesis date back to Robert Koch's identification of the tubercle bacillus in 1882 (149). Following inhalation, Mtb is engulfed by alveolar macrophages and initially confined within a membrane-bound phagosome (150). Rather than being eliminated, the bacterium actively disrupts phagosome maturation, blocking acidification and fusion with lysosomes (151). In parallel, Mtb promotes macrophage polarization towards an M2-like, anti-inflammatory state, thereby dampening microbicidal activity and facilitating bacterial persistence and immune evasion (152, 153).

Host cell variability strongly shapes these early intracellular events. Even a single bacterium may initiate systemic infection with highly different clinical outcomes (10). At the single-cell level, Rutschmann et al. showed that intracellular Mtb growth rates differ substantially between macrophages due to cell-to-cell variation in inducible nitric oxide synthase (iNOS) activity (154). This illustrates how stochastic immune effector variability, even within a nominally uniform macrophage population, directly shapes bacterial fate. Beyond iNOS, more host-cell factors contribute to variable infection outcomes. Single-cell studies demonstrate that subsets of alveolar macrophages are epigenetically and ontogenetically predisposed to either restrict or support bacterial replication (155, 156). Tissue context also shapes heterogeneity of outcomes: macrophages across the lung exhibit diverse metabolic states, phagolysosomal capacity, and inflammatory thresholds, all of which affect Mtb persistence. A recent genome-wide CRISPR screen identified components of the GID/CTLH E3 ubiquitin ligase complexes as key host factors that heterogeneously suppress antimicrobial responses in infected macrophages (157).

At the tissue scale, granulomas, structured aggregates of immune and stromal cells, represent a hallmark of TB pathology. Granulomas are highly heterogeneous, varying in cellular composition, immune activation, vascularization, metabolic microenvironment, and bacterial burden even within the same individual (158). Some granulomas effectively restrict bacterial replication and progress toward fibrosis, while others undergo necrosis and cavitation and promote bacterial dissemination through the airways (159). Single-cell studies and spatial transcriptomics have revealed that this diversity reflects underlying variation in both host immune responses and Mtb behavior within and across lesions (160–165).

Alongside host variability, Mtb bacterium exhibits extensive heterogeneity that fundamentally shapes infection outcomes. Distinct clinical lineages differ in immunomodulatory capacity, intracellular trafficking outcomes, and propensity for cytosolic escape (166). Even within a single strain, Mtb populations diversify into subpopulations that vary in growth rate, stress tolerance, and expression of regulons such as DosR, as well as in their degree of cell-wall lipid remodeling. These features influence persistence in macrophages and susceptibility to immune and antibiotic pressures (167). Mtb also occupies multiple host cell types like macrophages, epithelial cells, fibroblasts, and mesenchymal stem cells, with niche-specific adaptations that generate additional layers of heterogeneity (168). Although Mtb has also been detected in the host cell cytosol, the frequency, determinants, and functional implications of cytosolic escape remain incompletely understood (169).

Non-genetic heterogeneity in Mtb virulence is increasingly recognized as a driver of heterogenous infection outcomes across individuals and tissues. Time-lapse microscopy provides direct evidence that even daughter cells derived from a single Mtb mother cell diverge substantially in size, growth dynamics, and stress resistance, giving rise to subpopulations with distinct replication rates and antibiotic susceptibility (3). Furthermore, by exploiting its virulence systems, like the early secretory antigenic target secretion system-1 (ESX-1), Mtb promotes recruitment of specific macrophage subsets conducive to pathogen survival, thereby amplifying the heterogeneity of granuloma

formation and infection dynamics (165). Overall, these studies highlight that heterogeneity of Mtb and host cells jointly govern divergent infection outcomes across scales.

## CONTROL OF SINGLE-CELL INFECTION OUTCOMES: HOST VS. PATHOGEN

The combination of heterogeneous host and bacterial behavior leads to divergent infection outcomes at the single-cell level. What appears random or stochastic at the population level is often the result of these interacting, layered sources of variability. While many studies emphasize the single-cell nature of host-pathogen interactions and their functional consequences, the relative contributions of microbial-intrinsic factors versus host-driven determinants remain an area of active investigation.

Direct evidence for the co-variation between host and bacterial responses in relation to infection outcome was provided by Avraham et al., who investigated the activation of type I IFN responses in macrophages infected with *S. enterica*. They showed that bimodal expression of the two-component PhoP-PhoQ regulatory system in bacteria altered LPS composition on the surface of individual bacteria, resulting in heterogeneity of the IFN response in host macrophages. Therefore, the nature of the immune interactions was dictated by bacterial-intrinsic differences in bacterial virulence (13). In turn, Pisu et al. showed that the *in vivo* heterogeneity of heat shock protein X (hspX) stress responses to Mtb correlates with distinct macrophage transcriptional states, linked to bacterial dormancy and drug tolerance (155). Moran et al. employed a dual-color *L. monocytogenes* infection assay to track the fate of individual bacteria within single macrophages and thus assess both host and pathogen contributions to the infection outcome. By simultaneously infecting with genetically identical bacteria expressing either GFP or dsRed, they demonstrated that the ability of each bacterium to establish intracellular replication was statistically independent (8). When multiple bacteria entered the same host cell, they behaved non-cooperatively, acting independently from each other. This shows that the ability of *Listeria* to establish replicative infection in macrophages is driven by bacterial-intrinsic factors. However, the study did not exclude the possibility that bacterial fate may also be modulated by local variations in phagosome environment, such as differential acidification or membrane integrity at the bacteria-phagosome interface (170). In contrast, Rengarajan et al., using a sequential dual-color infection assay, demonstrated that host-derived variation can predetermine infection outcomes. They found that in vascular endothelial cells, the differences in susceptibility to bacterial invasion arose primarily from cell-to-cell variation in host cell surface adhesion. Notably, infection events were biased toward a subset of endothelial cells that repeatedly internalized bacteria, a pattern inconsistent with random entry. This enrichment stemmed from differences in host membrane properties controlling adherence. Notably, this adhesion-related susceptibility decayed rapidly, suggesting that infection likelihood is governed by transient functional states (109).

A complementary perspective was offered by Voznica et al., who also used dual-color sequential infections to understand epithelial susceptibility to *Salmonella* infection. Their approach distinguished between induced susceptibility, where an initial infection event increases the likelihood of subsequent infection, and inherent vulnerability due to pre-existing host traits. They found that susceptibility could be predicted from quantitative host cell features such as morphology, local crowding, or cholesterol content (140). Finally, also using a dual-color approach, Rutschman et al. demonstrated that two distinct Mtb strains display correlated growth when residing within the same macrophage (154). Nonetheless, replication rates varied among individual bacilli even withing a single host cell, indicating that bacterial heterogeneity also contributes to divergent intracellular outcomes. Together, these findings show that the dual-color infection assay is an effective tool to dissect the relative contributions of host- and pathogen-intrinsic heterogeneity to the outcome of the interaction.

The contribution of the host cell environment is also evident in macrophages, where *Salmonella* encounters a range of vacuolar environments. Upon phagocytosis, vacuolar acidification and nutrient stress activate the ppGpp- and Lon-dependent toxin-antitoxin

modules, driving phenotypic diversification: some bacteria begin replicating, while others enter a non-replicative persistent state (4). Notably, these persisters can resume intracellular growth after phagocytosis by naïve macrophages, suggesting that host heterogeneity diversifies environmental cues, but the bacterial program controls the switch to persistence. Similarly, the heterogeneity of granulomas and their ability to restrict Mtb arises from variations in the tissue environment. Oxygen level, vascularization, and the state of the recruited immune cells influence the granuloma structure and function independently of the pathogen (156, 160–162). This emphasizes the critical role of cell-intrinsic and microenvironmental factors determining infection outcomes and reinforces the concept of the spatial and phenotypic variability across host cell populations. Critically, the work of Voznica et al. highlights the broader potential of morphological and microenvironmental factors in predicting single-cell infection outcomes from spatially-resolved tissue-level data (140).

Growing evidence also points to the predictive power of extrinsic host states in determining infection outcomes. For example, Bossel Ben-Moshe et al. demonstrated that the distribution of white blood cell responses to *Salmonella* infection can be used to predict disease outcomes and stratify infection risk, even without spatial and microenvironmental information (171). These findings suggest that not only the variability of the pathogen or the spatial context of the infection but also the variability in host transcriptional and physiological states prior to infection significantly shape the nature of the immune response.

## MODELING HETEROGENEITY IN HOST-PATHOGEN INTERACTIONS: THE SYSTEMS BIOLOGY PERSPECTIVE

Systems biology and mathematical modeling offer a structured way to understand the heterogeneous nature of host-pathogen interactions and integrate mechanisms across scales.

### Modeling intracellular variability: stochastic gene expression and signaling dynamics

Pioneering theoretical approaches showed that modeling gene regulation as a stochastic process explains experimentally observed variability at mRNA and protein levels (172–174). Frameworks based on the chemical master equation (CME) and its simulation (e.g., via the Gillespie algorithm) formalized how molecular fluctuations arise and propagate when biomolecules (e.g., mRNAs and proteins) are present in low copy numbers (175, 176). Further advances came from the so-called telegraph models, which recapitulate transcriptional bursting by randomly switching between on and off promoter states in both host cells (177, 178) and bacteria (32). These models were subsequently extended to incorporate more complex regulation, for example, promoter cycling and combinatorial transcription factor binding (82, 179). Recent developments include the use of telegraph models for inference of global patterns from scRNA-seq data (80).

Mathematical models of bacterial virulence circuits increasingly show that phenotypic heterogeneity can arise from intrinsic stochasticity in regulatory networks. Models based on stochastic and nonlinear dynamics—often incorporating positive feedback, Hill-type cooperativity, or toxin-antitoxin relationships —successfully capture key features observed experimentally at the single-cell level, including the coexistence of high- and low-virulence states, the formation of persister subpopulations, and the asynchronous progression of invasion. These approaches have, for instance, been used to explain how bistable regulation of virulence genes in *Salmonella* gives rise to antibiotic-tolerant subpopulations (12), how stochastic activation of toxin-antitoxin modules, combined with environmental fluctuations, promotes persistence (180, 181), or how phenotypic noise can support bacterial cooperation behaviors during infection (139). Together, these models highlight that even clonal bacteria can diversify into functionally distinct states, providing a mechanistic explanation for the heterogeneous behaviors observed during infection.

Similar mathematical frameworks have been applied to innate immune signaling pathways involved in pathogen sensing, mainly NF-κB, and also STAT and IRF systems. Traditionally, these models involved the law of mass action to model deterministic population-level behavior (92, 182). The incorporation of stochastic description for biochemical reactions quantitatively reproduced heterogeneous digital activation, cell-to-cell variability in temporal responses, and all-or-none vs. graded responses observed in single-cell data, linking molecular noise directly to infection-relevant phenotypic differences (183–185). These models continue to be developed with ever-increasing complexity, allowing the generation of models of entire cells incorporating stochastic components (186). Recent developments involve the integration of transcriptional bursting models with signaling pathway models (81), explaining how complex signaling dynamics regulate stochastic gene expression patterns.

Although single-cell models of immune signaling exist, and separate models describe intracellular bacterial behavior, there is a limited mathematical framework that integrates both processes within the same individual host cell. As a result, the direct quantitative impact of signaling dynamics on single-bacterium fates remains largely unexplored.

## Agent-based and spatial models: scaling single-cell heterogeneity to tissue architecture

While intracellular stochasticity seeds heterogeneity, tissue-level infection outcomes are highly influenced by the spatial organization and interactions of diverse host and bacterial states. Agent-based models (ABMs) have been central to this understanding, with the most extensively validated multi-scale spatial framework in bacterial infection modeling emerging from tuberculosis research (187). In TB, ABMs, such as GranSim, simulate thousands of macrophages, T cells, and Mtb bacilli, each following experimentally informed rules, while cytokines, chemokines, and nutrients are represented as continuous fields (188). Hybrid and multiscale extensions embed these models in multi-compartment host frameworks, coupling cellular dynamics to cytokine signaling, pathogen metabolism, and oxygen and drug diffusion, thereby linking single-lesion activity to whole-organ and organism outcomes (189–192). Critically, these approaches also incorporate pathogen heterogeneity. Bacteria are modeled as discrete stochastic agents capable of variable replication, metabolic switching, virulence expression, and susceptibility to killing. Such variability is required to reproduce the coexistence of replicating, non-replicating, and hypoxia-adapted bacilli observed in vivo (188, 189).

Beyond tuberculosis, agent-based modeling has been applied to other intracellular bacterial pathogens, although to a lesser extent (187). In *Salmonella* infection, spatial ABMs of hepatic inflammation integrate immune cell recruitment, cytokine signaling, and bacterial dynamics to reproduce experimentally observed sepsis trajectories (193). Spatial models of human colon have also been used to investigate *Salmonella* colonization resistance and the microenvironmental influences on its invasion dynamics (194). For *L. monocytogenes*, Ortega et al. developed a stochastic ABM in which individual bacteria switch probabilistically between phenotypic states that differ in motility, protrusion formation, and junctional crossing efficiency, allowing rare *Listeria* transitions into highly motile cells to drive heterogeneous spatial expansions (114). In addition, ABMs have been applied to model *Listeria* persistence and spread in food processing environments (195), demonstrating the flexibility of agent-based frameworks to capture complex spatial and temporal dynamics across biological scales.

An emergent insight across the system-level models is the rare-event dominance, where rare bacterial or cellular events may disproportionately affect infection outcomes. ABMs and multiscale models provide mechanistic predictions suggesting which infection trajectories are probable, which are rare, and which may lead to negative patient outcomes, highlighting how single-cell variability scales into macroscopic outcomes such as granuloma stability, epithelial barrier breach, systemic dissemination, or even treatment failure (189–192). Dynamic optimization and optimal control theory further show how time-dependent interventions, such as antimicrobial dosing or host-directed

therapies, can be tuned to minimize persistence or dissemination (196). Importantly, these models allow *in silico* perturbation of parameters that are experimentally difficult to manipulate, revealing how altering the variability, and not only the mean immune response or virulence expression, can regulate infection outcomes. Such predictive insight may guide experimental design, helping us to identify and modulate cell states, regulatory motifs, and microenvironmental features that most likely determine infection fate through the control of cellular heterogeneity.

## SUMMARY AND FUTURE PERSPECTIVES

Altogether, existing studies demonstrate that single-cell infection outcomes emerge from dynamic, context-dependent interactions. At the cellular level, heterogeneity is driven by phenotypic variability in bacteria as well as inherent variability of host cells, which together elicit diverse transcriptional and functional responses. These responses, and ultimately pathogen fates, are further shaped by tissue-level regulation. In bacteria, the heterogeneity largely reflects intrinsic noise in transcriptional regulation, which ultimately may produce multistability and result in complex invasion strategies. In contrast, host cell heterogeneity arises from a combination of intrinsic and extrinsic factors, environmental inputs, as well as cellular memory, resulting in complex, multilayer regulation. To what extent bacterial- and host-driven mechanisms determine the outcomes depends on the stage of infection, the cell type involved, and the surrounding environment. Notably, rare events, such as infection with a single or small number of pathogens, can lead to different outcomes, as it has been shown for Mtb (10), through probabilistic effects on immune signaling, bacterial dissemination, and persistence. Dissecting these layers of regulation is essential to understanding how heterogeneous cellular responses collectively influence tissue-level pathogenesis and systemic disease progression. This apparently stochastic nature has important implications for both fundamental understanding and future therapies.

Mathematical modeling of infection dynamics and spatial simulations offers a promising approach that links local interaction and systemic effects by integrating different types of single-cell data (197). In parallel, development of standardized tools to measure heterogeneity across different levels and infection models is urgently needed to enable comparative analyses (198). Finally, advances in synthetic biology, including CRISPR-based tools, are required to precisely manipulate the variability of host and pathogen responses (68, 199, 200) and offer new approaches to probe the functional relevance of this heterogeneity in infection. A better understanding of how heterogeneity is generated and regulated across biological scales will support the development of new therapeutic strategies that could rely on modulation of this heterogeneity in a cell- and tissue-specific manner.

## ACKNOWLEDGMENTS

This work was supported by Polish National Agency for Academic Exchange (BPN/PPO/ 2022/1/00002) and National Science Centre Poland (2022/45/B/NZ6/01643).

The figure was created with BioRender.

## AUTHOR AFFILIATIONS

[1]Institute of Fundamental Technological Research, Polish Academy of Sciences, Warsaw, Poland
[2]University of Manchester, The Lydia Becker Institute of Immunology and Inflammation, Manchester, United Kingdom
[3]Division of Immunology, Immunity to Infection and Respiratory Medicine, Faculty of Biology, Medicine and Health, The University of Manchester, Manchester, United Kingdom

## AUTHOR ORCIDs

Juan Alfonso Redondo ⓘ http://orcid.org/0000-0002-9280-9848
Niharika Purkait ⓘ http://orcid.org/0009-0007-3780-2866
Pawel Paszek ⓘ http://orcid.org/0000-0002-0363-0716

## FUNDING

| Funder | Grant(s) | Author(s) |
| --- | --- | --- |
| Narodowa Agencja Wymiany Akademickiej | BPN/PPO/2022/1/00002 | Juan Alfonso Redondo |
| | | Pawel Paszek |
| Narodowe Centrum Nauki | 2022/45/B/NZ6/01643 | Niharika Purkait |
| | | Pawel Paszek |

## AUTHOR CONTRIBUTIONS

Juan Alfonso Redondo, Conceptualization, Visualization, Writing – original draft, Writing – review and editing | Niharika Purkait, Conceptualization, Writing – original draft, Writing – review and editing | Pawel Paszek, Conceptualization, Writing – original draft, Writing – review and editing

## ADDITIONAL FILES

The following material is available online.

Open Peer Review

**PEER REVIEW HISTORY (review-history.pdf).** An accounting of the reviewer comments and feedback.

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
