## [Reviewer comments · mSystems]

Heterogeneity of bacterial host-pathogen interactions across biological scales

Juan Redondo, Niharika Purkait, and Pawel Paszek

Corresponding Author(s): Pawel Paszek, Instytut Podstawowych Problemow Techniki Polskiej Akademii Nauk

Review Timeline:

Submission Date:	January 13, 2026
Editorial Decision:	February 22, 2026
Revision Received:	February 27, 2026
Accepted:	March 6, 2026

Editor: Yingying Pu

Reviewer(s): Disclosure of reviewer identity is with reference to reviewer comments included in decision letter(s). The following individuals involved in review of your submission have agreed to reveal their identity: Bingqing Li (Reviewer #1)

Transaction Report:

DOI: <https://doi.org/10.1128/msystems.01804-25>

Re: mSystems01804-25 (**Heterogeneity of bacterial host-pathogen interactions across biological scales**)

Dear Prof. Pawel Paszek:

Overall, the revised version has made substantial improvements. We note that Figure 1 and some sections have been either deleted or elaborated in more detail. Most importantly, the addition of a dedicated section discussing the mathematical modelling of heterogeneity in host-pathogen interactions from a systems biology perspective significantly enhances the completeness and depth of this review.

Before proceeding further, we would like you to consider a few remaining points from the reviewers:

The newly added section on mathematical modelling would benefit from a broader discussion beyond the current example. Expanding it to include other relevant pathogens would strengthen the comparative perspective and enhance the review's comprehensiveness.

Additionally, please carefully address the minor issues noted by the reviewers, including reference formatting, italicization of bacterial names, and the layout suggestion for Table 1.

Revision Guidelines

Sincerely,
Yingying Pu
Editor
mSystems

Reviewer #1 (Comments for the Author):

The authors have thoughtfully addressed all comments and suggestions raised in the previous round of review. The revisions are comprehensive and have significantly improved the clarity, coherence, and scholarly rigor of the manuscript.

Reviewer #2 (Comments for the Author):

Overall, the revised version has made substantial improvements. Specifically, Figure 1 and some sections have been either deleted or elaborated in more detail. Most importantly, a dedicated section has been added to discuss the mathematical modelling of heterogeneity in host-pathogen interactions from a systems biology perspective, which significantly enhances the completeness and depth of this review.

Major Comments

1. In the new section on the mathematical modelling, the discussion mainly focuses on *Mycobacterium tuberculosis* as an example. The review would be more comprehensive if similar discussions on *Listeria* and *Salmonella* were also included.

Minor Comments

1. Check and standardize the reference numbers in lines 103 and 109.
2. The bacterial name in line 374 should be italicized. Please check the entire manuscript for such errors.
3. Consider using a horizontal layout for Table 1 to improve readability and better present the table content. (for reference only, not mandatory)

Overall, the revised version has made substantial improvements. Specifically, Figure 1 and some sections have been either deleted or elaborated in more detail. Most importantly, a dedicated section has been added to discuss the mathematical modelling of heterogeneity in host-pathogen interactions from a systems biology perspective, which significantly enhances the completeness and depth of this review.

Major Comments

1. In the new section on the mathematical modelling, the discussion mainly focuses on *Mycobacterium tuberculosis* as an example. The review would be more comprehensive if similar discussions on *Listeria* and *Salmonella* were also included.

Minor Comments

1. Check and standardize the reference numbers in lines 103 and 109.
2. The bacterial name in line 374 should be italicized. Please check the entire manuscript for such errors.
3. Consider using a horizontal layout for Table 1 to improve readability and better present the table content. (for reference only, not mandatory)

Reviewer #2 (Comments for the Author):

Overall, the revised version has made substantial improvements. Specifically, Figure 1 and some sections have been either deleted or elaborated in more detail. Most importantly, a dedicated section has been added to discuss the mathematical modelling of heterogeneity in host-pathogen interactions from a systems biology perspective, which significantly enhances the completeness and depth of this review.

Thank you for these comments. Please see below for our responses. Changes in the revised manuscript are highlighted using TrackChanges.

Major Comments

1. In the new section on the mathematical modelling, the discussion mainly focuses on *Mycobacterium tuberculosis* as an example. The review would be more comprehensive if similar discussions on *Listeria* and *Salmonella* were also included.

We have now extended the relevant section, emphasizing that tuberculosis provides the most extensive and well-validated examples of complex infection modelling, while highlighting additional applications for *Salmonella* and *Listeria*:

Lines 570-573: Agent-based models (ABMs) have been central to this understanding, with the most extensively validated multi-scale spatial framework in bacterial infection modelling emerging from tuberculosis research [187].

Lines 587-598: Beyond tuberculosis, agent-based modelling has been applied to other intracellular bacterial pathogens, although to lesser extent [187]. In *Salmonella* infection, spatial ABMs of hepatic inflammation integrate immune cell recruitment, cytokine signaling and bacterial dynamics to reproduce experimentally observed sepsis trajectories [193]. Spatial models of human colon have also been used to investigate *Salmonella* colonization resistance and the microenvironmental influences on its invasion dynamics [194]. For *L. monocytogenes*, Ortega *et al.* developed a stochastic ABM in which individual bacteria switch probabilistically between phenotypic states that differ in motility, protrusion formation and junctional crossing efficiency, allowing rare *Listeria* transitions into highly motile cells to drive heterogenous spatial expansions [114]. In addition, ABMs have been applied to model *Listeria* persistence

and spread in food processing environments [195], demonstrating the flexibility of agent-based frameworks to capture complex spatial and temporal dynamics across biological scales.

Minor Comments

1. Check and standardize the reference numbers in lines 103 and 109.

These and other references have been checked and corrected throughout the manuscript. One additional reference by Reffsin et al, from Jan 2026, demonstrating that single cell susceptibility to viral infection is driven by variable cell state, was added (DOI: [10.1016/j.cell.2025.10.021](https://doi.org/10.1016/j.cell.2025.10.021), ref 6, line 56 in the revised manuscript).

2. The bacterial name in line 374 should be italicized. Please check the entire manuscript for such errors.

This and other instances have been corrected.

3. Consider using a horizontal layout for Table 1 to improve readability and better present the table content. (for reference only, not mandatory).

We have now presented the table in a landscape format.

Re: mSystems01804-25R1 (**Heterogeneity of bacterial host-pathogen interactions across biological scales**)

Dear Prof. Pawel Paszek:

Your manuscript has been accepted, and I am forwarding it to the ASM production staff for publication. Your paper will first be checked to make sure all elements meet the technical requirements. ASM staff will contact you if anything needs to be revised before copyediting and production can begin. Otherwise, you will be notified when your proofs are ready to be viewed.

Sincerely,
Yingying Pu
Editor
mSystems

Reviewer #2 (Comments for the Author):

The revised manuscript has addressed all my comments. I have no further concerns.